# Assessment of tuberculosis transmission probability in three Thai prisons based on five dynamic models

Nithinan Mahawan[1], Thanapoom Rattananupong[1], Puchong Sri-Uam[2], Wiroj Jiamjarasrangsi[1]*

**1** Department of Preventive and Social Medicine, Faculty of Medicine, Chulalongkorn University, Bangkok, Thailand, **2** Center for Safety, Health and Environment of Chulalongkorn University, Bangkok, Thailand

* wjiamja@gmail.com

**Data Availability Statement:** All protocol and dataset files are available from the protocols.io database (https://www.protocols.io/view/assessment-of-tuberculosis-transmission-

## Abstract

This study aimed to assess and compare the probability of tuberculosis (TB) transmission based on five dynamic models: the Wells–Riley equation, two Rudnick & Milton-proposed models based on air changes per hour (ACH) and liters per second per person (L/s/p), the model proposed by Issarow *et al*, and the Applied Susceptible-Exposed-Infected-Recovered (SEIR) TB transmission model. This study also aimed to determine the impact of model parameters on such probabilities in three Thai prisons. A cross-sectional study was conducted using data from 985 prison cells. The TB transmission probability for each cell was calculated using parameters relevant to the specific model formula, and the magnitude of the model agreement was examined by Spearman's rank correlation and Bland–Altman plot. Subsequently, a multiple linear regression analysis was conducted to investigate the influence of each model parameter on the estimated probability. Results revealed that the median (Quartiles 1 and 3) of TB transmission probability among these cells was 0.052 (0.017, 0.180). Compared with the pioneered Wells–Riley's model, the remaining models projected discrepant TB transmission probability from less to more commensurate to the degree of model modification from the pioneered model as follows: Rudnick & Milton (ACH), Issarow *et al.*, and Rudnick & Milton (L/s/p), and the applied SEIR models. The ventilation rate and number of infectious TB patients in each cell or zone had the greatest impact on the estimated TB transmission probability in most models. Additionally, the number of inmates in each cell, the area per person in square meters, and the inmate turnover rate were identified as high-impact parameters in the applied SEIR model. All stakeholders must urgently address these influential parameters to reduce TB transmission in prisons. Moreover, further studies are required to determine their relative validity in accurately predicting TB incidence in prison settings.

## Introduction

Tuberculosis (TB) is the leading cause of illness and death from communicable diseases worldwide [1]. This is also applicable to Thailand, which is one of 30 countries with the highest TB

probabilit-6qpvr868zlmk/v1) (DOI: dx.doi.org/10.17504/protocols.io.6qpvr868zlmk/v1) and the Figshare repository (DOI: 10.6084/m9.figshare.26089744).

**Funding:** - WJ and NM received the 90th Anniversary of Chulalongkorn University Fund (grant no. 035, 1/2021) and the FY2021 Thesis Grant for Doctoral Degree Study of the National Research Council of Thailand (NRCT; grant no. N41D640002, 2021). - URI (the 90th Anniversary of Chulalongkorn University Fund): https://www.grad.chula.ac.th/index.php?lang=th -URI: https://www.nrct.go.th/ -No, the funders had no role in study design, data collection and analysis, decision to publish, or preparation of the manuscript.

**Competing interests:** The authors have declared that no competing interest exist.

burden [1]. Although global TB incidence declined by approximately 9% from 2015 to 2019, the challenge of high TB incidence among high-risk populations, such as prisoners, where TB incidence rates are 5 to 70 times higher than in the general population, still persists [2, 3]. As Thailand has the 7[th] highest number of prisoners (incarcerated individuals) in the world [4], TB incidence in prisons is of particular public health concern in its attempt to meet the World Health Organization's (WHO) End TB Strategy milestone [3].

High TB risk among prisoners is caused by environmental factors, such as overcrowding, limited access to healthcare, insufficient access to nutrition, poor ventilation, indoor and unsanitary confinement, disruption of social and support networks, and frequent movement within the carceral system. Additionally, personal risk factors, such as HIV infection status, cigarette smoking, illegal drug use, and limited access to healthcare before incarceration, contribute to the increased risk of TB among prisoners [5]. However, the prison environment, rather than host risk factors, is the main driver of TB risk [5]. This is supported by an observational and modeling study in Brazil that found that the incidence of latent TB was low at the time of incarceration, increased during incarceration, and declined after release to that of the general population over a seven-year period [6]. To reduce TB incidence among incarcerated individuals, specific aspects of the prison environment should therefore be targeted. Since prisons often act as institutional amplifiers of TB, implementing ameliorating measures will result in less TB spillover into the general population [6].

Currently, a few dynamic models of TB transmission have been applied in prison settings based on prison environmental features. These models include the classical Wells–Riley model [7] and its variant models with a progressive degree of modification, namely the Rudnick & Milton-proposed models [8, 9], the Issarow *et al.*-proposed model [10, 11], and the Applied Susceptible-Exposed-Infected-Recovered (SEIR) TB transmission model [12]. The Wells–Riley model and its three modified models similarly approach the airborne transmission of infectious disease as a physical transport problem, that is, how an organism gets from one human host to another [13], with some minor differences. Both Wells–Riley's and Rudnick & Milton's models quantify the infectious dose based on the concept of "quantum of infection," although the former assumes steady-state ventilation conditions and the latter does not [7–9]. However, Issarow *et al.*'s models are further modified by quantifying the infectious dose based on the infectors' average volume fraction of exhaled air using carbon dioxide level as a surrogate marker of exhaled air as well as respiratory deposition fraction, and assuming either steady-state or nonsteady-state conditions [10, 11]. Conversely, the SEIR model is an epidemiological model that comprises compartments representing sets of individuals grouped by disease status. The link between compartments represents transitions from one disease state to another, and the TB transmission probability can be estimated by determining the model's basic reproductive number [14].

Each of these models predicts the probability of TB transmission based on different prison environmental factors, although some factors are shared by them all. All of these models have the potential to identify prison environmental features with high leverage as intervention targets to reduce TB transmission. As these models have been separately utilized in previous studies, their comparability or discrepancy in projecting the risk of TB transmission is yet to be investigated.

This study aimed to (a) assess the probability of TB transmission in three Thai prisons using the five previously mentioned dynamic models; (b) examine the agreement among the TB transmission probability assessed in (a); and (c) determine the impact of model parameters, such as prison architectural and environmental features, TB management, and inmates' demographics and TB-related characteristics on TB transmission probability.

## Materials and methods

### Study population

A cross-sectional study was conducted in three Thai prisons, with all prison cells for permanent residences: one supermaximum prison (Prison A, with five zones, 239 cells, and 4,734 inmates), one maximum prison (Prison B, with six zones, 652 cells, and 6,607 inmates) in Bangkok, and one maximum prison (Prison C, with four zones, 94 cells, and 6,216 inmates) in Rayong province, approximately 180 kilometers east of Bangkok. Each cell serves as the study and analysis unit, and a total of 985 cells from these prisons were included in this study. The study received ethical approval from the Ethical Review Board of Chulalongkorn University Faculty of Medicine, with reference number 610/63 and gained permission from the Department of Corrections at the Ministry of Justice before being conducted. Additionally, the superintendent of each correctional facility reviewed the "Information Sheet for Participants" and signed the "Submission Agreement for Volunteers" before initiating the study. No personal information was collected during the study that could identify inmates; thus, individual inmate consent was not required.

### Data collection

The following variables were used to calculate the probability of TB transmission for each cell: (1) Cell architectural and environmental characteristics, such as cell ventilation, room volume, outdoor air exchange rate, opening facing prevailing winds, opening into the building, and number of cells per courtyard; (2) the demographic and health status composition of inmates, including the number of total and susceptible inmates per cell, residence time per day, and inmate turnover; (3) TB infection- and progression-related parameters, such as the number of infectious patients, infectious quanta rate, surviving airborne infectious doses, transmission rate, proportion of fast progressors, rate of fast progressors developing infectious TB (primary progression), rate of fast progressors moving to slow progressors, short-term latently infected, rate of slow progressors developing infectious TB (reactivation), long-term latently infected, relapse rate, partial acquired immunity after primary infection for treated persons, rate of recovery under antituberculosis treatment, and natural recovery; (4) physiological parameters, including pulmonary ventilation rate and respiratory deposition fraction of airborne infectious particles; and (5) treatment effectiveness parameters, including period of infectiousness, number of recovered patients, and TB-related, natural, and other mortality rates. These variables and parameters were collected or obtained as shown in S1 Table.

### Cell architectural and environmental characteristics

For security reasons, five inmates from each zone were recruited and trained by the principal investigator (NM) and a coinvestigator (PS, who is also an industrial hygienist) to survey the architectural and environmental characteristics of each cell within the zone. The data collected included floor area (m$^2$) and ceiling height (m) to calculate cell volume (V; m$^3$), size of open windows and doors (m$^2$), number of cells per courtyard, presence of cross-ventilation design (yes or no), temperature (°C), relative humidity (%), and the number and size of portable fans. In addition, the trained inmate team measured the area of the courtyard (m$^2$) and the outdoor wind speed (meters/second).

### Ventilation rate

As in previous studies, two methods were used to measure the cell ventilation rate in the real world, including the absolute ventilation rate (liters/second/person) and air changes per hour

(ACH) [7, 8, 10, 12]. In Method 1, the absolute ventilation rate, always used as a surrogate for exhaled air, was assessed by steady-state carbon dioxide ($CO_2$) in parts per million using the Kimo HQ210 with SCOH 112 probe (Sauermann Industries, ZA Bernard Moulinet, Montpon, France). The $CO_2$ concentrations in the morning after at least 13 hours of lockup time within and outside each cell were measured by the trained inmate team [8–10]. The absolute ventilation rate (Q; L/s) for each cell was then calculated based on the equilibrium between ventilation and $CO_2$ production through respiration, according to Persily's model [15]. In Method 2, ACH (Q; ACH), as a rule of thumb, is classically used as a metric for assessing infection control risk. It is the total air volume in a room or space that is completely removed and replaced in an hour [16]. In this method, the wind speed (meters/second) in each cell was measured at the location of the opening facing prevailing winds using a hot wire thermo-anemometer and datalogger (Model SDL350, Extech Instruments, Waltham, MA). The credibility of the data was cooperatively ensured by the principal investigator (NM) and a coinvestigator (PS). Wind speed data was then combined with data on cell volume and the size of openings facing prevailing winds to calculate the ACH according to the WHO-proposed model [16].

## Inmate demographics and health status

Data on the number of all and susceptible inmates with HIV or chronic disease and aged 60 or older (S; in persons), residence time per day (θ or t; in hours), and inmate turnover rate (Ω; in percent per year) were obtained from primary data sources, such as surveys or interviews with staff and inmates, and secondary data sources, such as reports or computer databases corresponding to each zone in the prisons.

## Physiological parameters

Data on the adult pulmonary ventilation rate (p; in liters/hour or $m^3$/hour) [17, 18] and the deposition fraction of airborne infectious particles that successfully reach the target infectious site of the host respiratory system (θ; in percent per surviving airborne infectious doses) were obtained from published studies [19].

## TB infection- and progression-related parameters

Data provided comprised the number of tuberculosis-infected patients in each cell and zone, which served as an indication of the prevalence at both the cell and zonal levels. These data were considered secondary data, comprising documents and computer databases sourced from the respective zonal and central administrative and medical facilities within the prisons. The TB diagnosis was based on the 9th and 10th Revisions of the International Classification of Diseases (ICD 9&10) codes A15.0 (Tuberculosis of lung confirmed by sputum microscopy with or without culture), A15.1 (Tuberculosis of lung confirmed by culture only), and A16.0 (Tuberculosis of lung, bacteriologically and histologically negative). Collection of these data commenced in October 2020. The data was recorded as a percentage for a period of 6 months, from July to December 2020, for Prisons A and B, and from October 2020 to March 2021 for Prison C. The TB infection- and progression-related parameters were obtained from relevant international literature. These parameters included data on the quanta of infectious particles produced per hour (q; in quanta/hour) [7, 8, 20–25], surviving airborne infectious doses per hour (β-μ; in doses $hr^{-1}$) [11], the proportion of fast progressors (p; in percent) [26, 27], the rate of fast progressors developing infectious TB ($\sigma_1$; per year) [26, 28], the rate of fast progressors moving to slow progressors ($\sigma_2$; per year) [26, 29], the rate of slow progressors developing infectious TB (ω; per year) [30], the relapse rate (r; per year) [30], partial acquired immunity after primary infection for treated persons (*f*; percent) [26, 28], the rate of recovery under

antituberculosis treatment ($\alpha$; percent) [12, 31], and the natural recovery rate ($\alpha_n$; per year) [30, 32]. However, some TB infection-and progression-related parameters relied on data from three Thai prisons. These parameters included the TB transmission rate ($\beta$; per person per year), calculated using the formula PQ/VA, where P is the average pulmonary ventilation rate (0.36 m$^3$/hour), Q is the quanta production rate per infector (12.7 quanta/hour), V is the room volume per inmate (m$^3$), and A is the ventilation rate (ACH) [33]; the numbers of short-term latently infected (L1; in persons per year), calculated using the formula S$\beta$I, where S is the number of susceptible inmates, $\beta$ is the TB transmission rate, and I is the number of TB-infectious patients in each cell and zone (prevalence); and long-term infected individuals (L2; in persons per year), calculated using the formula (1-p) L1 $\sigma_2$, where p is the proportion of fast progressors [26, 27], L1 is the number of short-term latently infected, and $\sigma_2$ is the rate of fast progressors moving to slow progressors [26, 29].

## Treatment effectiveness parameters

According to previous studies, the average period of infectivity or time exposure prior to diagnosis is 180 days [34, 35]. In this study, however, the average infectivity period (t; in days) or "time-to-TB diagnosis" from the studied prisons' database (i.e., inferring from the time-lapse from the date of annual chest x-ray TB surveillance, which differed for each zone of the prison, to the date of TB treatment initiated for each diagnosed inmate) was calculated. The number of recovered patients (R; in percent per year) and natural/other mortality rates ($\mu$; in percent per year) were also obtained from secondary data collected in three Thai prisons. In addition, data on the TB-related death rate ($\mu_1$; in percent per year) was obtained from the published study [30, 36].

## TB transmission probability modeling

The following five dynamic models were used to calculate the probability of TB transmission: Wells–Riley's equation [7], two Rudnick & Milton's models based on ACH and liters/second/person (L/s/p) as ventilation rate parameters, respectively [8, 9], the SEIR TB transmission model [12], and Issarow et al.'s model [10, 11] (S2 Table).

## Statistical analysis

Cells were divided into three to six categories based on architectural and environmental characteristics, primarily using quartiles with modifications for meaningful and mutually exclusive categories as appropriate to ensure sufficient samples in each category. For categorical variables, the data were summarized as frequencies and percentages, while medians and interquartile ranges (IQR) were used for continuous variables. The subgroups were compared using appropriate statistical tests. For categorical data, the chi-squared test or Fisher's exact test was used when more than 20% of cells had expected frequencies less than 5. However, nonparametric tests such as the Wilcoxon rank–sum (Mann–Whitney) test were used for continuous, non-normally distributed data.

Agreement among the TB transmission probability estimated using the different dynamic models was then assessed via Spearman's rank correlation ($\rho$). The coefficient values were interpreted for the degree of agreement as weak ($<0.50$), moderate (0.50–0.69), strong (0.70–0.89), and very strong (0.90–1.0) [37]. The detailed pattern of agreement or difference was further investigated using Bland–Altman plots representing absolute and percent differences [38]. To investigate the influence of model parameters on TB transmission probability, two procedures were used separately for each TB transmission prediction model. First, cells were categorized into four subgroups based on the quartile of predicted TB transmission

probability. Furthermore, subgroups of model parameters were then compared using the Wilcoxon rank–sum (Mann–Whitney) test since the cell-specific probabilities of TB transmission were non-normally distributed. Second, the magnitude and pattern of change in the predicted TB transmission probability along each model's parameter categories were investigated using a multiple linear regression incorporating all model parameters simultaneously for a specific prediction model. The adjusted beta (i.e., the magnitude of change in the predicted TB transmission probability) and its 95% confidence intervals were estimated. All statistical analyses were performed using Stata software, Version 13.0 (StataCorp. 2013, Stata Statistical Software: Release 13, College Station, TX: StataCorp LP).

## Results

### Cell characteristics

In terms of the architectural characteristics of the two prisons (Prisons A and B) in Bangkok (the capital city), cells in Prison A had a larger average floor area (28.00 *vs*. 7.71 m$^2$), a higher ceiling (3.60 *vs*. 3.48 m), and a much larger cell volume (100.80 *vs*. 26.98 m$^3$) than those in Prison B (Table 1). However, cells in Rayong province's Prison C had a much larger average floor area (41.36 m$^2$) but a lower ceiling height (2.80 m) than those in Bangkok's prisons, with a larger cell volume (115.81 m$^3$). In terms of ventilation (Table 1), cells in Prison C had the highest average ventilation rates (39.93 ACH and 33.99 L/s/p), while cells in Prison A and B had comparable ventilation rates (29.81 *vs*. 28.93 ACH and 24.41 *vs*. 24.53 L/s/p).

The average number of inmates in cells was highest in Prison C (55 persons), followed by Prison A (23 persons), and lowest in Prison B (six persons), with a corresponding reverse pattern of area per person (2.16 *vs*. 4.38 *vs*. 5.20 m$^3$/person). The cell inmate turnover rates were highest in Prison C (40.30% per year), much lower in Prison A (8.40% per year), and lowest in Prison B (1.69% per year).

During the past six months, there was at least one TB case in 39 out of 239 cells (16.32%) in Prison A, 55 out of 652 cells (8.44%) in Prison B, and 35 out of 94 cells (37.23%) in Prison C. However, when considering the presence of TB cases in the zone as a potential source of TB exposure for inmates in each cell, the prevalence of infectious patients in the zone during the past six months was highest in Prison B (1.32%), followed by Prison A (0.98%), and Prison C (0.90%) (Table 1). Furthermore, the average time-to-TB diagnosis, calculated as the time from the date of getting ill to the date of formal TB diagnosis, was the longest in Prison A (365.21 days), followed by Prison C (161.93 days) and Prison B (116.39 days).

### TB transmission probability

Fig 1 illustrates the estimated TB transmission probability within cells using five prediction models. While the probability values varied obviously, the relative patterns of TB transmission probabilities among the three prisons were mostly comparable to those predicted by the Wells–Riley and the two Rudnick & Milton-proposed models. In particular, the highest transmission probabilities were observed in Prison A, while Prisons C and B had lower probabilities. The median TB transmission probabilities in the three prisons, as estimated by the Wells–Riley model, ranged from 0.036 to 0.100, while those estimated by the two Rudnick & Milton-proposed models ranged from 0.040 to 0.135 and 0.019 to 0.044. However, the models proposed by Issarow *et al*. and the applied SEIR models predicted different TB transmission probabilities from other models, both in terms of values and relative patterns. Notably, the applied SEIR model estimated TB transmission probabilities ranging from 0.157 to 0.570, with the highest probabilities observed in Prison C.

**Table 1. Cell architectural, environmental, and tuberculosis risk characteristics in the three prisons studied.**

| Local parameter | Formula | | | | | Prison A ($n$ = 239) | | Prison B ($n$ = 652) | | $p$ | Prison C ($n$ = 94) | | $p$ |
|---|---|---|---|---|---|---|---|---|---|---|---|---|---|
| | M1 | M2 | M3 | M4 | M5 | Med | (Q1, Q3) | Med | (Q1, Q3) | | Med | (Q1, Q3) | |
| **Architectural characteristics** | | | | | | | | | | | | | |
| Floor area (m$^2$) † | - | - | - | - | - | 28.00 | (28.00, 28.00) | 7.71 | (7.71, 16.44) | a | 41.36 | (41.36, 41.36) | a, b |
| Ceiling height (m) † | - | - | - | - | - | 3.60 | (3.60, 3.60) | 3.48 | (3.35, 3.50) | a | 2.80 | (2.80, 2.80) | a, b |
| Cell volume (m$^3$) † | - | √ | √ | - | - | 100.80 | (100.80, 100.80) | 26.98 | (25.82, 57.22) | a | 115.81 | (115.81, 115.81) | a, b |
| **Percent of the opening area in walls** | | | | | | | | | | | | | |
| To prevailing wind (%) † | - | - | - | - | - | 3.93 | (3.93, 3.93) | 2.99 | (2.60, 3.11) | a | 2.05 | (2.05, 3.95) | a, b |
| To building (%) † | - | - | - | - | - | 15.18 | (15.18, 15.18) | 1.05 | (1.00, 4.02) | a | 5.67 | (0, 5.67) | a, b |
| Number of ventilation fans† | - | - | - | - | - | 1 | (1, 1) | 1 | (0, 1) | a | 0 | (0, 1) | a, b |
| Ventilation rate (ACH) † | √ | √ | - | √ | √ | 29.81 | (23.40, 39.25) | 28.93 | (17.24, 46.86) | - | 39.93 | (21.52, 64.59) | a, b |
| Absolute ventilation rate (L/s/p) † | - | - | √ | - | - | 24.41 | (18.57, 35.14) | 24.53 | (19.19, 30.86) | - | 33.99 | (24.64, 48.6) | a, b |
| **Demographics and health status of cell inmates** | | | | | | | | | | | | | |
| Number of inmates in the cell † | √ | √ | √ | √ | √ | 23 | (13, 25) | 6 | (5, 11) | a | 55 | (46, 60) | a, b |
| Area per person (m$^3$/person) † | - | - | - | √ | - | 4.38 | (4.03, 5.60) | 5.20 | (4.77, 5.63) | a | 2.16 | (1.93, 2.52) | a, b |
| Inmate turnover rate (%/year) † | - | - | - | √ | - | 8.40 | (5.69, 9.54) | 1.69 | (1.39, 6.86) | a | 40.30 | (35.11, 41.50) | a, b |
| Time to diagnosis TB cases in the cell (days)† | √ | √ | √ | - | √ | 365.21 | (298.56, 446.53) | 116.39 | (70.85, 116.39) | a | 161.93 | (151.81, 161.93) | a, b |
| Number of overall TB cases in the cell (prevalence in 180 days) [n (%)] ‡ | √ | - | - | √ | - | | | | | a | | | a, b |
| 0 | | | | | | 200 | (83.68) | 597 | (91.56) | | 59 | (62.77) | |
| 1 | | | | | | 30 | (12.55) | 20 | (3.07) | | 20 | (21.28) | |
| > 1 | | | | | | 9 | (3.77) | 35 | (5.37) | | 15 | (15.96) | |
| The prevalence of infectious patients in the cell (prevalence per 180 days) † | √ | - | - | √ | - | 0 | (0, 0) | 0 | (0, 0) | a | 0 | (0, 0.02) | a, b |
| The prevalence of infectious patients in the zone (prevalence per 100 persons per 180 days) † | √ | - | - | √ | - | 0.98 | (0.53, 1.72) | 1.32 | (0.49, 1.32) | - | 0.90 | (0.57, 1.12) | - |
| Number of TB cases in the cell by smear status [n (%)] ‡ | - | √ | √ | - | √ | | | | | a | | | a, b |
| No | | | | | | 200 | (83.68) | 597 | (91.56) | | 59 | (62.77) | |
| Yes, but smear-negative | | | | | | 21 | (8.79) | 20 | (3.07) | | 0 | (0) | |
| Yes, and smear-positive | | | | | | 11 | (4.60) | 2 | (0.31) | | 35 | (37.23) | |
| Yes, and both smear-negative and smear-positive | | | | | | 7 | (2.93) | 33 | (5.06) | | 0 | (0) | |
| The prevalence of infectious patients in the cell by smear status (prevalence per 180 days) † | - | √ | √ | - | √ | | | | | | | | |
| Smear-negative | | | | | | 0 | (0, 0) | 0 | (0, 0) | - | 0 | (0, 0) | a, b |
| Smear-positive | | | | | | 0 | (0, 0) | 0 | (0, 0) | - | 0 | (0, 0.02) | a, b |
| The prevalence of infectious patients in the zone by smear status (prevalence per 100 persons per 180 days) † | - | √ | √ | - | √ | | | | | | | | |
| Smear-negative | | | | | | 0.42 | (0.32, 0.91) | 0.82 | (0.30, 0.82) | - | 0 | (0, 0) | a, b |

*(Continued)*

**Table 1.** (*Continued*)

| Local parameter | Formula | | | | | Prison A (*n* = 239) | | Prison B (*n* = 652) | | *p* | Prison C (*n* = 94) | | *p* |
|---|---|---|---|---|---|---|---|---|---|---|---|---|---|
| | M1 | M2 | M3 | M4 | M5 | Med | (Q1, Q3) | Med | (Q1, Q3) | | Med | (Q1, Q3) | |
| **Smear-positive** | | | | | | 0.44 | (0.21, 0.76) | 0.51 | (0.20, 0.51) | - | 0.90 | (0.57, 1.20) | a, b |

M1, Wells–Riley model [7]; M2 and M3, Rudnick & Milton-proposed models (M2, ventilation rate in air changes per hour (ACH); M3, ventilation rate in liters/second/person (L/s/p)) [8, 9]; M4, the Applied Susceptible-Exposed-Infected-Recovered (SEIR) tuberculosis transmission model [12]; M5, the models proposed by Issarow *et al*. [10, 11]; Med (Q1, Q3), median (Quartile 1, Quartile 3)

[a]Statistical significance levels of less than 0.05 upon comparing Prisons A and B and Prisons A and C.

[b] Statistical significance levels of less than 0.05 upon comparing Prisons B and C.

†A two-sample Wilcoxon rank–sum (Mann–Whitney) test was used to compare medians.

‡ A chi-squared or Fisher's exact test.

## Agreement among the five dynamic models

Fig 2 illustrates the correlations of TB transmission probability among the dynamic models. Inferring the classical Wells–Riley's model as the origin, Milton & Rudnick's (ACH) model demonstrated the highest and very strong correlation ($\rho$ = 0.90) with Wells–Riley's model, followed by Issarow *et al*.'s ($\rho$ = 0.78) and Milton & Rudnick's (L/s/p) ($\rho$ = 0.72) models. Conversely, the applied SEIR model indicated only moderate correlation ($\rho$ = 0.57) with Wells–Riley's model and weak to moderate correlation ($\rho$ = 0.33–0.57) with the remaining models. Further investigation using the Bland–Altman plot revealed that transmission probabilities

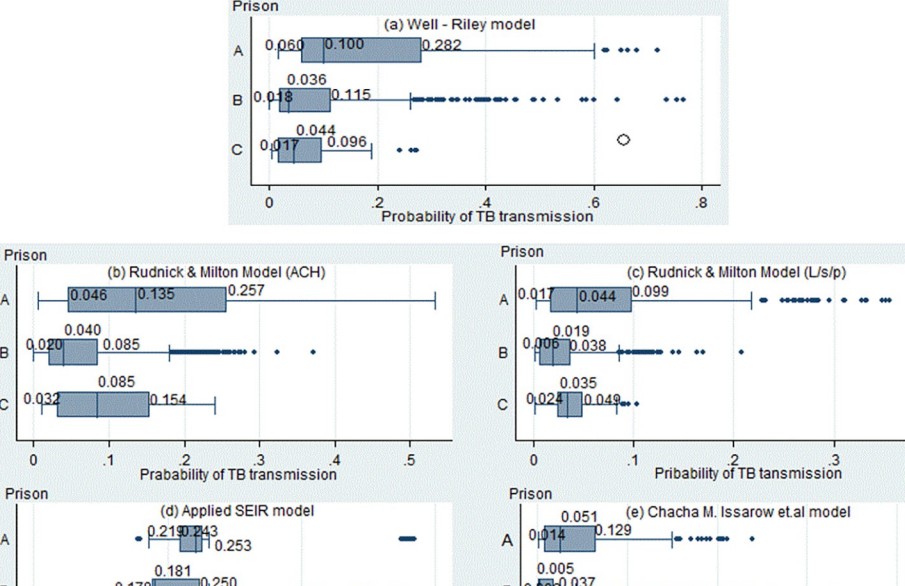

**Fig 1. The estimated probability of tuberculosis transmission using five prediction models.**

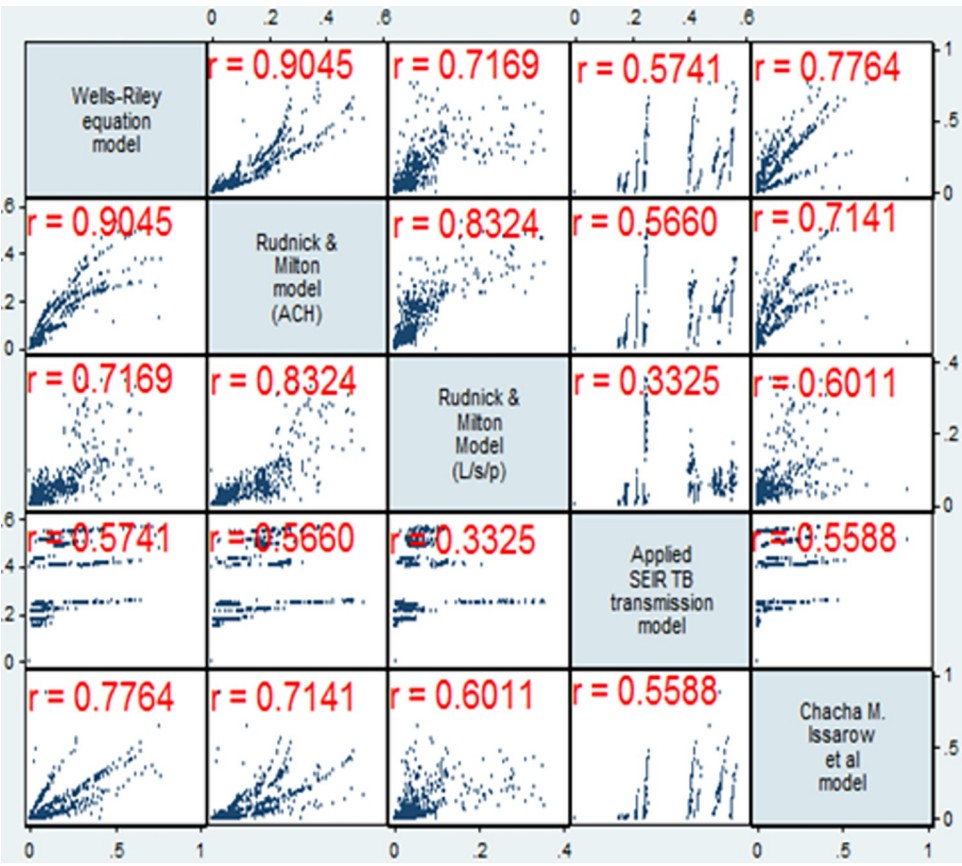

**Fig 2. Spearman-rank correlations (ρ) between the five prediction models ($n = 985$, $p < 0.001$ for all pairs).**

estimated by nearly all models were higher than that of Wells–Riley's model, with the average magnitude varying along the probability level from +70% to +80% for the Rudnick & Milton (ACH) model (S1(A) and S1(B) Fig), from +90% to +150% for the Rudnick & Milton (L/s/p) model (S1(C) and S1(D) Fig); and from +110% to +180% for the Issarow *et al.* model (S1(E) and S1(F) Fig). The exception was the applied SEIR model, of which the estimated probability was lower than that of the Wells–Riley model at the lower probability range and comparable or higher at the higher probability range, with the average magnitude from −100% to +20% (S1(G) and S1(H) Fig).

## Relationship between the model parameters and predicted probability of TB transmission

The role of each parameter in differentiating the TB transmission probability was inferred from its progressive increase or decrease (i.e., dose–response pattern) based on the quartile of the TB transmission probability. The results showed that all parameters used in all models demonstrated a perfect or nearly perfect dose–response pattern (Table 2), particularly in the Rudnick & Milton (L/s/p) model, where four out of eight parameters demonstrated such property. The most distinct parameters showing a perfect dose–response pattern were "time-to-TB diagnosis" in four out of four models, "prevalence of TB cases in cell/zone: overall, smear-positive, and smear-negative TB cases together" in five out of five models, and "number of inmates in the cell showing such property" in five out of five models.

**Table 2. Quartiles of the cell-specific probability of tuberculosis transmission.**

| Local parameter | Quartiles of tuberculosis transmission probability | | | | | | | | | | |
|---|---|---|---|---|---|---|---|---|---|---|---|
| | **Q1** | | **Q2** | | *p*‡ | **Q3** | | *p*‡ | **Q4** | | *p*‡ |
| | Med | (Q1, Q3) | Med | (Q1, Q3) | | Med | (Q1, Q3) | | Med | (Q1, Q3) | |
| **The Wells–Riley model** | | | | | | | | | | | |
| (TB transmission probability) | (<0.0252) | | (0.0252–0.0486) | | | (0.0487–0.1536) | | | (> 0.1536) | | |
| **Number of inmates in the cell** | 5.00 | (5.00, 5.00) | 6.00 | (5.00, 12.00) | a | 23.00 | (8.00, 26.00) | a | 23.00 | (11.00, 27.00) | ab |
| **Number of overall TB cases in the cell §** | 0.50 | (0.50, 0.50) | 1.32 | (0.50, 1.32) | a | 0.98 | (0.50, 1.32) | ab | 2.15 | (1.80, 2.15) | abc |
| **Prevalence of overall TB cases in the cell** | 0 | (0, 0) | 0 | (0, 0) | a | 0 | (0, 0) | ab | 0 | (0, 0.03) | abc |
| **Prevalence of overall TB cases in the zone** | 0.50 | (0.50, 0.50) | 1.32 | (0.50, 1.32) | a | 0.98 | (0.50, 1.32) | ab | 2.13 | (1.72, 2.15) | abc |
| **Ventilation rate (ACH)** | 38.83 | (26.99, 60.14) | 48.31 | (38.25, 56.45) | a | 24.67 | (18.49, 31.10) | ab | 15.92 | (12.93, 24.86) | abc |
| **Time-to-TB diagnosis in the cell (months)** | 2.36 | (2.36, 2.36) | 3.88 | (3.88, 5.50) | a | 5.50 | (3.88, 9.95) | ab | 6.88 | (3.88, 14.88) | abc |
| **The Rudnick & Milton-proposed models (ACH)** | | | | | | | | | | | |
| (TB transmission probability) | (<0.0240) | | (0.0240–0.0480) | | | (0.0481–0.1590) | | | (> 0.1590) | | |
| **Number of inmates in the cell** | 5 | (5, 9) | 6 | (5, 12) | a | 22 | (6, 32) | ab | 23 | (11, 26) | ab |
| **Number of smear-negative patients in the cell §** | 0.30 | (0.30, 0.30) | 0.42 | (0.32, 0.81) | a | 0.33 | (0.22, 0.81) | a | 1.16 | (0.91, 1.64) | abc |
| **Number of smear-positive patients in the cell §** | 0.20 | (0.20, 0.20) | 0.51 | (0.17, 0.51) | a | 0.51 | (0.44, 0.89) | ab | 0.98 | (0.80, 1.02) | abc |
| **Prevalence of smear-negative patients in the cell** | 0 | (0, 0) | 0 | (0, 0) | - | 0 | (0, 0) | ab | 0 | (0, 0) | abc |
| **Prevalence of smear-positive patients in the cell** | 0 | (0, 0) | 0 | (0, 0) | - | 0 | (0, 0) | ab | 0 | (0, 0) | ab |
| **Prevalence of smear-negative patients in the zone** | 0.30 | (0.30, 0.30) | 0.42 | (0.32, 0.82) | a | 0.33 | (0.22, 0.82) | a | 1.16 | (0.91, 1.64) | abc |
| **Prevalence of smear-positive patients in the zone** | 0.20 | (0.20, 0.20) | 0.51 | (0.17, 0.51) | a | 0.51 | (0.44, 0.89) | ab | 0.98 | (0.80, 0.98) | abc |
| **Ventilation rate (ACH)** | 37.44 | (27.69, 52.34) | 48.58 | (22.37, 59.08) | - | 29.23 | 1 (9.21, 38.61) | ab | 17.22 | 1 (3.72, 25.22) | abc |
| **Time-to-TB diagnosis in the cell (days)** | 70.85 | (70.85, 121.45) | 116.39 | (116.39, 164.97) | a | 151.81 | 1 (16.39, 253.02) | ab | 206.42 | 1 (16.39, 446.53) | abc |
| **Cell volume (m³)** | 26.98 | (25.82, 55.08) | 26.98 | (26.98, 57.22) | a | 100.80 | (26.98, 115.81) | ab | 100.80 | (57.22, 113.67) | abc |
| **The Rudnick & Milton-proposed models (L/s/p)** | | | | | | | | | | | |
| (TB transmission probability) | (<0.0080) | | (0.0080–0.0250) | | | (0.0251–0.0500) | | | (> 0.0500) | | |
| **Number of inmates in the cell** | 5 | (5, 5) | 11 | (5, 23) | a | 11 | (6, 29) | ab | 24 | (11, 30) | abc |
| **Number of smear-negative patients in the cell §** | 0.30 | (0.30, 0.30) | 0.33 | (0.33, 0.81) | a | 0.81 | (0.22, 0.81) | ab | 1.16 | (0.91, 1.64) | abc |
| **Number of smear-positive patients in the cell §** | 0.20 | (0.20, 0.20) | 0.21 | (0.17, 0.51) | a | 0.51 | (0.51, 0.90) | ab | 0.89 | (0.76, 0.98) | abc |
| **Prevalence of smear-negative patients in the cell** | 0 | (0, 0) | 0 | (0, 0) | a | 0 | (0, 0) | a | 0 | (0, 0) | abc |
| **Prevalence of smear-positive patients in the cell** | 0 | (0, 0) | 0 | (0, 0) | a | 0 | (0, 0) | ab | 0 | (0, 0) | ab |
| **Prevalence of smear-negative patients in the zone** | 0.30 | (0.30, 0.30) | 0.33 | (0.33, 0.82) | a | 0.82 | (0.22, 0.82) | ab | 1.16 | (0.91, 1.64) | abc |
| **Prevalence of smear-positive patients in the zone** | 0.20 | (0.20, 0.20) | 0.21 | (0.17, 0.51) | a | 0.51 | (0.51, 0.90) | ab | 0.89 | (0.76, 0.98) | abc |
| **Ventilation rate (L/s/p)** | 31.90 | (26.53, 39.69) | 26.80 | (20.39, 35.14) | a | 23.01 | (19.55, 29.21) | ab | 19.12 | (15.25, 24.53) | abc |
| **Time-to-TB diagnosis in the cell (days)** | 70.85 | (70.85, 70.85) | 164.97 | (116.39, 164.97) | a | 116.39 | (116.39, 161.93) | ab | 206.42 | (116.39, 446.53) | abc |
| **Cell volume (m³)** | 26.98 | (25.82, 26.98) | 55.08 | (25.82, 100.80) | a | 57.22 | (26.98, 115.81) | ab | 100.80 | (57.22, 115.81) | abc |
| **The Applied Susceptible-Exposed-Infected-Recovered (SEIR) tuberculosis transmission model** | | | | | | | | | | | |
| (TB transmission probability) | (<0.1783) | | (0.1783–0.2202) | | | (0.2203–0.4105) | | | (> 0.4105) | | |
| **Number of inmates in the cell** | 5 | (5, 5) | 6 | (5, 6) | a | 22.5 | (12, 25) | ab | 25 | (11, 49) | abc |
| **Number of overall TB cases in the cell §** | 0.49 | (0.49, 0.49) | 1.32 | (1.32, 1.32) | a | 0.53 | (0.50, 1.72) | ab | 2.12 | (1.20, 2.15) | abc |
| **Prevalence of overall TB cases in the cell** | 0 | (0, 0) | 0 | (0, 0) | a | 0 | (0, 0) | ab | 0 | (0, 0.02) | abc |
| **Prevalence of overall TB cases in the zone** | 0.49 | (0.49, 0.49) | 1.32 | (1.32, 1.32) | a | 0.53 | (0.50, 1.72) | ab | 2.08 | (1.20, 2.15) | abc |
| **Ventilation rate (ACH)** | 34.26 | (25.87, 45.42) | 48.30 | (34.51, 57.51) | a | 22.73 | (16.86, 31.44) | ab | 18.17 | (13.83, 36.82) | abc |
| **Area per person (m³/person)** | 5.40 | (5.16, 5.40) | 5.16 | (4.50, 5.40) | a | 4.77 | (4.38, 5.60) | a | 4.58 | (2.36, 5.51) | abc |
| **Inmate turnover rate (%/year)** | 1.39 | (1.39, 1.39) | 1.69 | (1.41, 1.69) | a | 8.40 | (4.64, 9.66) | ab | 20.79 | (8.38, 37.36) | abc |

*(Continued)*

**Table 2.** (Continued)

| Local parameter | Quartiles of tuberculosis transmission probability | | | | | | | | | |
|---|---|---|---|---|---|---|---|---|---|---|
| | Q1 | | Q2 | | *p*‡ | Q3 | | *p*‡ | Q4 | | *p*‡ |
| | Med | (Q1, Q3) | Med | (Q1, Q3) | | Med | (Q1, Q3) | | Med | (Q1, Q3) | |
| **Transmission rate (persons/year)** | 0.025 | (0.017, 0.035) | 0.019 | (0.015, 0.028) | a | 0.041 | (0.030, 0.055) | ab | 0.054 | (0.041, 0.079) | abc |
| **The models proposed by Issarow *et al.*** | | | | | | | | | | |
| **(TB transmission probability)** | (<0.0030) | | (0.0030–0.0110) | | | (0.0111–0.0750) | | | (> 0.0750) | | |
| **Number of inmates in the cell** | 5 | (5, 5) | 6 | (5, 9) | a | 12 | (10, 24) | ab | 28 | (24, 42) | abc |
| **Number of smear-negative patients in the cell §** | 0.30 | (0.30, 0.30) | 0.81 | (0.33, 0.81) | a | 0.33 | (0.32, 1.16) | a | 0.95 | (0.00, 1.64) | abc |
| **Number of smear-positive patients in the cell §** | 0.20 | (0.20, 0.20) | 0.51 | (0.17, 0.51) | a | 0.51 | (0.21, 0.98) | ab | 0.90 | (0.81, 1.20) | abc |
| **Prevalence of smear-negative patients in the cell** | 0 | (0, 0) | 0 | (0, 0) | a | 0 | (0, 0) | a | 0 | (0, 0) | abc |
| **Prevalence of smear-positive patients in the cell** | 0 | (0, 0) | 0 | (0, 0) | a | 0 | (0, 0) | a | 0 | (0, 0.1) | abc |
| **Prevalence of smear-negative patients in the zone** | 0.30 | (0.30, 0.30) | 0.82 | (0.33, 0.82) | a | 0.33 | (0.32, 1.16) | a | 0.91 | (0.00, 1.64) | abc |
| **Prevalence of smear-positive patients in the zone** | 0.20 | (0.20, 0.20) | 0.51 | (0.17, 0.51) | a | 0.51 | (0.21, 0.98) | ab | 0.89 | (0.81, 1.20) | abc |
| **Ventilation rate (ACH)** | 36.64 | (26.81, 50.71) | 43.40 | (30.10, 55.66) | a | 22.93 | (15.29, 37.10) | ab | 20.08 | (13.86, 29.22) | abc |
| **Time-to-TB diagnosis in the cell (months)** | 2.36 | (2.36, 2.36) | 3.88 | (3.88, 5.50) | a | 5.50 | (3.88, 9.95) | ab | 6.88 | (5.06, 14.88) | ab |

Med (Q1, Q3), median (Quartile1, Quartile 3)

‡ A two-sample Wilcoxon rank–sum (Mann–Whitney) test was used to compare medians.

a Statistical significance levels of less than 0.05 upon comparing Q1 and 2, Q1 and 3, and Q1 and 4.

b Statistical significance levels of less than 0.05 upon comparing Q2 and 3 and Q2 and 4.

c Statistical significance levels of less than 0.05 upon comparing Q3 and 4.

§ Sum of the prevalence of infectious patients in the cell (per 180 days) and in the zone (per 100 persons per 180 days).

Furthermore, the impact of parameter changes on altering the TB transmission probability was investigated using multiple linear regression analysis. The results revealed that the magnitudes of altered probability were generally the highest across all the prediction models, particularly with categorical changes in ventilation of 4%–20% points and the number of TB cases in the zone of 1%–23% points, depending on the prediction model (S3 Table). According to the applied SEIR model, a marked alteration in TB transmission probability was observed for categorical changes in the "number of inmates in cell" by 4%–12% points, the "area per person in square meter" by 11%–20% points, and the "inmate turnover rate" by 7%–22% points.

## Discussion

In this study, five predictive models, i.e., the Wells–Riley model, two Rudnick and Milton-proposed models based on ACH and ventilation rate parameters (L/s/p), models proposed by Issarow *et al.*, and the applied SEIR model, were used to assess the probabilities of TB transmission in cells of three large Thai prisons. The results revealed that the median (Quartiles 1 and 3) of TB transmission probability among these cells was 0.052 (0.017, 0.180). Compared with the classical Wells–Riley's model, Rudnick & Milton's (ACH) model demonstrated the highest agreement, followed by Rudnick & Milton's (L/s/p) and Issarow *et al.*'s models, and SEIR model showing lowest agreement. Further analysis revealed that the "ventilation rate" and the "number of infectious TB patients in the cell/zone" had the greatest impact on the estimated TB transmission probability in most models. However, the "number of inmates in the cell," the "area per person in square meters," and the "inmate turnover rate" were identified as high-impact parameters in the applied SEIR model.

## TB transmission probability in prisons

Magnitude of TB transmission probability reported in our study differed from previous studies. Four previous studies utilized dynamic models to investigate TB transmission probability in prison settings. The first two studies, by Johnstone–Robertson *et al*. explored the interactions between incarceration conditions (including cell ventilation, lock-up time, and TB incidences and treatment delays) and TB control measures in a South African prison, whereas Cooper–Arnold *et al*. investigated the impact of potential risk factors, specifically indoor ventilation, on TB infection among deputy sheriffs during an outbreak in a short-term urban lockup in Connecticut, USA. Using the Wells–Riley's model, Johnstone–Robertson *et al*. reported a TB transmission probability of 5.4%–90% [7], and Cooper–Arnold *et al*. reported 0%–15% [39]. Conversely, our study revealed a TB transmission probability of 0%–77%. These discrepancies may be attributed to the higher and wider range of ventilation rates and time-to-TB diagnosis revealed in our study compared with Johnstone–Robertson *et al*.'s study (0–194 *vs*. 1–12 ACH and 71–507 *vs*. 0–180 days, respectively) [7]. However, Cooper–Arnold *et al*. reported ventilation rates of 1763 and 4954 cubic feet per minute for measuring and designing, respectively, or 22.48–63.19 L/s/p compared with 8.64–577.78 L/s/p in our study, and an exposure time of 19.1 hours per 3 days [39].

Urrego *et al*. conducted a study to assess the effect of ventilation and early diagnosis on TB transmission in Brazilian prisons using the Rudnick and Milton-proposed model and reported a TB transmission probability ranging from 14.2% to 99.9% in three prisons [8]. However, our study showed a TB transmission probability ranging from 0% to 53.3% in three prisons, when the ventilation rate was assessed in ACH, and 0.01% to 35.5%, for absolute ventilation rate (L/s/p). These discrepancies may be attributed to the lower level of ventilation rate reported in their study.

The last study was conducted by Naning *et al*. to simulate the impact of different TB treatment interventions against improved environmental conditions on the probability of TB transmission in an overcrowded Malaysian prison using the SEIR model. They reported a TB transmission probability ranging from 0.83% to 8.80% [12], compared to 15.65% to 56.97% in our study. This discrepancy may be attributed to our reported higher inmate turnover rate (0.28%–392.62% per year) versus Naning *et al*.'s study (33%–240% per year). However, no studies utilizing the models proposed by Issarow *et al*. have been reported in prison settings so far.

## Agreement among the dynamic models

Inferring from Spearman's rank correlation and Bland–Altman plot, result revealed that the variant models proposed by Rudnick & Milton and Issarow *et al*. had resulted in significantly higher estimates of TB transmission probability yet maintained high correlation with the pioneered Wells–Riley's model. This was the opposite of the applied SEIR model that estimated lower TB transmission probability and demonstrated only weak to moderate correlation with Wells–Riley's model and the other three models. These results were consistent with the degree of modification of each variant model. The discrepancy of the applied SEIR model may be attributed to the different theoretical concepts of the SEIR model compared with the other four models.

The lesser agreement between Issarow *et al*.'s and Wells–Riley's ($\rho$ = 0.78 and +110%–+180% difference) models than that between Rudnick & Milton's (Q in ACH) and Wells–Riley's ($\rho$ 0.91 and +70–+80% difference) models confirms the greater departure of the former model from the original Wells–Riley's model than the latter model. This verifies the significant effect of the extensive modifications made in Issarow *et al*.'s model compared with the

Rudnick & Milton's models (Q in ACH). However, the agreement between Rudnick & Milton's model (L/s/p) and Wells–Riley's model ($\rho = 0.72$ and +90%–+150% difference) markedly decreased when a different type of ventilation rate was used, as in the Rudnick & Milton's model (L/s/p),. This may be due to the low agreement between the two types of ventilation rate measures (i.e., ACH *vs*. L/s/p), with a $\rho$ value of only 0.33.

However, based on the previously mentioned evidence, the relative validity of these dynamic models could not be determined. However, they must be determined solely by their ability to predict TB incidence, which will be the focus of our upcoming report.

## Environmental factors having high impact on TB transmission probability in prison

Concerning the impactful parameters and interventions on TB transmission probability in prison settings, previous simulation studies revealed that these parameters included ventilation [7, 8, 12, 39], time-to-TB diagnosis [8], level of overcrowding [7, 12], active case-finding with increased ventilation and decreased lockup time [7, 40], and an appropriate treatment strategy with reduced overcrowding and increased ventilation [12, 28]. Among these, the "ventilation rate" and "active case-finding together with other measures" were most frequently reported as high-impact parameters or interventions [40, 41]. In our study, all parameters used in the five prediction models were found to contribute significantly to the assessment of TB transmission probability. However, only the "ventilation rate" and "number of infectious TB patients in the cell/zone" were shown to have a high impact on the predicted probability. Our finding of the "ventilation rate" as the highest-impact parameter in most predicting models aligns well with the existing evidence. In addition, our finding of the "number of infectious TB patients in the cell/zone" as a high-impact parameter is implicitly consistent with the prevailing evidence on the impact of "active case-finding" and "appropriate treatment strategy." These measures, together with other measures, can reduce the number of active TB cases in the cell/zone. Therefore, these two parameters can be considered the most effective levers for designing interventions aimed at reducing TB transmission probability in prison settings. Importantly, our findings were based on actual data rather than simulations, including low-transmission-risk cells that serve as desirable local examples. This makes our findings practical and applicable in real-world settings, particularly within the context of Thai prisons.

In 2021, the reported average occupancy rates of the three studied prisons were 136.44%, 120%, and 145.40, implying that the average cell floor area is less than the WHO recommendation of 5.40 m$^2$ floor area per inmate [42]. Our finding of "number of inmates in the cell," "area per person in square meters," and "inmate turnover rate" as high-impact parameters only in the applied SEIR model is consistent with previous evidence about the "level of overcrowding" reported by Johnstone–Robertson *et al*. [7] and Naning *et al*. [12]. A high "inmate turnover rate" indicates that the number of TB-exposed inmates increased, leading to an increase in the number of TB-infected inmates. However, while a previous study [8] found "time-to-TB diagnosis" to be a high-impact parameter for TB transmission probability, such a finding was not observed in ours. This discrepancy may be due to our determining of the "time-to-TB diagnosis" parameter indirectly from health administrative data at the zone level of prisons rather than directly from each individual TB inmate, resulting in the inaccuracy of this parameter data.

## Strengths and limitations

This study has some advantages. First, it included five dynamic models, allowing for simultaneous examination and comparison of these models. Second, it had a large sample size (number of cells) and covered three large prisons with different architectural and administrative

characteristics and geographical locations. This ensures sufficient statistical power and generalizability of the study results. However, it has some limitations. First, the "time-to-TB diagnosis" parameter was derived indirectly from health administrative data at the prison zone level rather than directly from each individual TB inmate. Second, some TB progression parameters were obtained from the published literature rather than locally from the Thai population. These two limitations may have led to inaccuracies in estimating the TB transmission probability and evaluating the impact of these parameters on such a probability. Finally, the coronavirus disease 2019 pandemic that occurred during the study period may have affected some model parameters related to prison and health management. The inmate health details were obtained from incomplete secondary data stored in each prison zone. Therefore, we did not use this data for analysis. Consequently, in this study, we defined all inmates in the sample cell as susceptible individuals.Therefore, the estimate of TB transmission probability may deviate from what would be expected during normal times. The pandemic, however, only occurred when the field data collection was almost complete.

## Conclusion

This study showed that the variant models projected different values of TB transmission probability. Although the probability values differed, three models, i.e., the Wells–Riley model and the two Rudnick & Milton-proposed models, estimated similar patterns of TB transmission probability. Using the pioneered Wells–Riley's model as the reference, the remaining models projected discrepant TB transmission probability from less to more commensurate to the degree of model modification from the pioneered model as follows: Rudnick & Milton (ACH), Issarow *et al*. and Rudnick & Milton (L/s/p), and the applied SEIR models. In terms of risk factors, our study identified two parameters that significantly contribute to ongoing TB transmission risk in all models: low ventilation rates and a high number of existing TB inmates in the cell or zone. All stakeholders must urgently address these issues to reduce TB transmission in prisons. Furthermore, since these five models produced varying estimates of TB transmission probabilities, further studies are required to determine their relative validity in accurately predicting TB incidence in prison settings.

## Supporting information

**S1 Fig. Bland–Altman plots portray the agreement pattern among the dynamic models.**
(DOCX)

**S1 Table. Parameters used in prediction models.**
(DOCX)

**S2 Table. Models used in predicting the probability of tuberculosis transmission.**
(DOCX)

**S3 Table. Multiple linear regression analysis of the association between model parameters and changes in tuberculosis transmission probability predicted using five prediction models (n = 985).**
(DOCX)

## Acknowledgments

The investigators would like to thank all stakeholders in the three prisons in this study for their assistance in data collection and survey. The investigators wish to thank the Department of Corrections at the Ministry of Justice, which gave permission for the study.

## Author Contributions

**Conceptualization:** Nithinan Mahawan, Wiroj Jiamjarasrangsi.

**Data curation:** Nithinan Mahawan, Wiroj Jiamjarasrangsi.

**Formal analysis:** Nithinan Mahawan, Thanapoom Rattananupong, Puchong Sri-Uam, Wiroj Jiamjarasrangsi.

**Funding acquisition:** Nithinan Mahawan, Wiroj Jiamjarasrangsi.

**Investigation:** Nithinan Mahawan, Puchong Sri-Uam, Wiroj Jiamjarasrangsi.

**Methodology:** Nithinan Mahawan, Thanapoom Rattananupong, Puchong Sri-Uam, Wiroj Jiamjarasrangsi.

**Project administration:** Nithinan Mahawan, Wiroj Jiamjarasrangsi.

**Resources:** Nithinan Mahawan, Puchong Sri-Uam, Wiroj Jiamjarasrangsi.

**Software:** Nithinan Mahawan, Thanapoom Rattananupong, Wiroj Jiamjarasrangsi.

**Supervision:** Nithinan Mahawan, Thanapoom Rattananupong, Puchong Sri-Uam, Wiroj Jiamjarasrangsi.

**Validation:** Nithinan Mahawan, Thanapoom Rattananupong, Puchong Sri-Uam, Wiroj Jiamjarasrangsi.

**Visualization:** Nithinan Mahawan, Wiroj Jiamjarasrangsi.

**Writing – original draft:** Nithinan Mahawan, Wiroj Jiamjarasrangsi.

**Writing – review & editing:** Nithinan Mahawan, Thanapoom Rattananupong, Puchong Sri-Uam, Wiroj Jiamjarasrangsi.

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
