## [Decision Letter · Decision Letter 0]

26 Mar 2024

PONE-D-24-02877Assessment of tuberculosis transmission probability in three Thai prisons based on five dynamic modelsPLOS ONE

Dear Dr. Jiamjarasrangsi,

Thank you for submitting your manuscript to PLOS ONE. After careful consideration, we feel that it has merit but does not fully meet PLOS ONE’s publication criteria as it currently stands. Therefore, we invite you to submit a revised version of the manuscript that addresses the points raised during the review process.

However, this is quite a relevant topic and so comprehensive study, well designed and written. It is also so detailed for analysis that has been undertaken in the prison setup regarding TB transmission in Thai.

However, the abstract section was not well summarized.  The discussion section should have summarized the study key finding and the comparator studies should also be briefly summarized, rather than listing out their study findings in number/percentage/CI as it is, e.g lines 336-339.  

Also revise the conclusion section for the study which should have summarized and interpreted the findings with future recommendation. The conclusion section looks like a discussion.

We look forward to receiving your revised manuscript.

Kind regards,

Zewdu Gashu Dememew, M.D, PhD

Academic Editor

PLOS ONE

2. In the online submission form, you indicated that your data will be submitted to a repository upon acceptance.  We strongly recommend all authors deposit their data before acceptance, as the process can be lengthy and hold up publication timelines. Please note that, though access restrictions are acceptable now, your entire minimal dataset will need to be made freely accessible if your manuscript is accepted for publication. This policy applies to all data except where public deposition would breach compliance with the protocol approved by your research ethics board. If you are unable to adhere to our open data policy, please kindly revise your statement to explain your reasoning and we will seek the editor's input on an exemption.

Additional Editor Comments:

Dear Authors,

This is quite a relevant topic and so comprehensive study, well designed and written. It is also so detailed for analysis that has been undertaken in the prison setup regarding TB transmission in Thai.

However, the abstract section was not well summarized. The discussion section should have summarized the study key finding and the comparator studies should also be briefly summarized, rather than listing out their study findings in number/percentage/CI as it is, e.g lines 336-339.

Also revise the conclusion section for the study which should have summarized and interpreted the findings with future recommendation. The conclusion section looks like a discussion.

Reviewers' comments:

Reviewer's Responses to Questions

**Comments to the Author**

1. Is the manuscript technically sound, and do the data support the conclusions?

Reviewer #1: Yes

Reviewer #2: Partly

2. Has the statistical analysis been performed appropriately and rigorously? 

Reviewer #1: Yes

Reviewer #2: I Don't Know

3. Have the authors made all data underlying the findings in their manuscript fully available?

Reviewer #1: Yes

Reviewer #2: No

4. Is the manuscript presented in an intelligible fashion and written in standard English?

Reviewer #1: Yes

Reviewer #2: No

5. Review Comments to the Author

Reviewer #1: The study provides valuable insights into the assessment of tuberculosis transmission probability in Thai prisons using dynamic models.

The authors have appropriately discussed the impact of model parameters on TB transmission probability and identified key influential factors.

It is commendable that the study highlights the need for urgent attention to these influential parameters to reduce TB transmission in prisons.

The authors suggest further studies to determine the relative validity of these parameters in accurately predicting TB incidence in prison settings.

The financial disclosure and competing interests have been appropriately addressed and declared by the authors.

The ethics statement provides necessary information regarding the approval and review of the study.

The data availability statement explains the limitations on public data sharing due to security reasons and provides information on accessing confidential data through the Ethics Committee.

Reviewer #2: The estimation of TB transmission probabilities is a crucial component of the planning of TB programs. However, the manuscript has several limitations.

Major comments

Certain variables have be part of the data collection or at least obtained in similar settings. This includes for pulmonary ventilation rate, the relapse rate, partial acquired immunity after primary infection for treated persons, the rate of recovery under antituberculosis treatment, and TB-related death rate. Inferring those data from published study may not be a valid input data.

Define "susceptible inmates" and state where the information about their health was obtained and how it was done.

It is necessary to specify the TB case definition, including the diagnosis methods.

The incidence of TB at Zone and cell was inferred using data obtained from documents and computer databases. Does the influence of under diagnosed and under reporting is considered?

The time-to-TB diagnosis is determined using the time the individual diagnosed by x-ray to treatment initiated, but the required time is the time exposure prior to diagnosis. Please clarify.

Include the details of how the number of recovered patients and mortality rates is determined. Does this information utilized as input data? A summary of the input data used for each model is more informative.

The variation in input parameters may account for the differences in the median TB transmission probability among the models. What is the rationale to utilize the different model with varied input parameters?

Minor comments

An introduction that includes an overview of the model's description; justification for applying each the model i.e an explanation of its benefits and limitations.

The average occupancy rates have to be move from result section to the discussion.

Discus about the key message for the TB program in the prison settings.

6. PLOS authors have the option to publish the peer review history of their article (what does this mean?). If published, this will include your full peer review and any attached files.

Reviewer #1: **Yes: **NUHA AMER Al-Aghbari

Reviewer #2: No

---

## [Author Response · Author response to Decision Letter 0]

17 May 2024

Dear academic editor and reviewers, 

We are grateful for the opportunity to revise our work on ‘Assessment of tuberculosis transmission probability in three Thai prisons based on five dynamic models ". We would to thank all the Reviewers and editor for your excellent comments and suggestions. We also appreciate your suggestions on revising in the abstract, introduction, discussion and conclusion, which were immensely helpful for revising the manuscript. 

We hope that our responses adequately address these comments and substantially improve the quality of our manuscript. We have included the reviewer comments responded to you individually, indicating exactly how we addressed each concern or problem and describing the changes we have made (detail on Response to Reviewers file). 

We hope the revised manuscript will better suit the PLOS ONE but are happy to consider further acceptation, and we thank you for your continued interest in our research. 

Yours Sincerely, 

Wiroj Jiamjarasrangsi, MD, PhD. 

Corresponding Author

---

## [Editor Report · Decision Letter 1]

28 May 2024

Assessment of tuberculosis transmission probability in three Thai prisons based on five dynamic models

PONE-D-24-02877R1

Dear Dr. Jiamjarasrangsi,

We’re pleased to inform you that your manuscript has been judged scientifically suitable for publication and will be formally accepted for publication once it meets all outstanding technical requirements.

Kind regards,

Zewdu Gashu Dememew, M.D, PhD

Academic Editor

PLOS ONE

---

## [Editor Report · Acceptance letter]

11 Jul 2024

PONE-D-24-02877R1 

PLOS ONE

Dear Dr. Jiamjarasrangsi, 

I'm pleased to inform you that your manuscript has been deemed suitable for publication in PLOS ONE. Congratulations! Your manuscript is now being handed over to our production team.

Kind regards, 

on behalf of

Dr. Zewdu Gashu Dememew 

Academic Editor

PLOS ONE